# Response Mechanism of Farmers' Livelihood Capital to the Compensation for Rural Homestead Withdrawal—Empirical Evidence from Xuzhou City, China

**Weiyan Qi [1], Zhemin Li [2],\* and Changbin Yin [1],\***

[1] Institute of Agricultural Resources and Regional Planning, Chinese Academy of Agricultural Sciences, Beijing 100081, China
[2] Agricultural Information Institute, Chinese Academy of Agricultural Sciences, Beijing 100081, China
\* Correspondence: lizhemin@caas.cn (Z.L.); yinchangbin@caas.cn (C.Y.)

**Abstract:** The Chinese government has implemented a homestead withdrawal policy to improve the efficiency of rural construction land use. The compensation for rural homestead withdrawal (CRHW) is crucial to the reconstruction and sustainable development of farmers' livelihoods. This paper analyzed the response mechanisms of farmers' livelihoods to the CRHW with the combined application of the logistic regression, the mediation effect model, and the moderating effect model. The results indicated that CRHW had a significant positive impact on the sustainable livelihoods of rural households, mainly by improving the physical capital and social capital. In addition, adaptability and livelihood diversity played intermediary and regulatory roles in the positive impacts of the CRHW on sustainable livelihoods, respectively. The conclusions may provide insight into the demand for more reasonable compensation policies to ensure the sustainability of farmers' livelihoods.

**Keywords:** compensation; rural homestead withdrawal; sustainable livelihoods; livelihood capital; adaptability; livelihood diversity

## 1. Introduction

China is a typical urban–rural dual structure society where urban and rural lands are independent (regarding management and utilization) [1]. Rural residents can acquire farmland and homesteads free of charge by virtue of their collective memberships [2,3]. Farmers use the former to engage in agricultural production to maintain their standards of living, and use the latter to build houses to ensure their basic livelihoods. Therefore, homesteads and farmhouses are the most important and valuable livelihood assets for China's rural residents. However, due to the rapid urbanization and industrialization in the last few decades, people have moved from rural to urban and, further, to regional centers and large coastal cities, resulting in a shortage of urban construction land, and idle, wasteful, and inefficient utilization of rural homesteads [4–6]. Since 2000, the annual scale of vacant rural houses has grown to 594 million square meters due to the rural population transfer [7]; the vacancy rate of rural homesteads is 20% [8]. According to China's national conditions (regarding a large population and limited land), improving the utilization efficiency of rural homesteads is a realistic demand to optimize the allocation of urban and rural land resources, and promote urban–rural integrated development. The Chinese government has implemented a rural land reform policy known as withdrawal from rural homestead(s) (WRH), which encourages farmers to give up their occupied homesteads (including their houses and other attachments) voluntarily and relocate to urban or rural concentrated settlements after obtaining reasonable compensation [9].

Since the implementation of the WRH policy, the withdrawal compensation for rural homestead withdrawal (CRHW) has been the core factor for policy evaluations [10], CRHW is related to the realization of farmers' land property rights and is simultaneously

involved in the maintenance of rural residents' long-term livelihoods. Rural homesteads have value, i.e., housing security, social security, and ecological security, as well as property/emotional functions [11–13]. These are rights enjoyed by farmers (as individuals and collective members) [14]. At present, compensation for farmers who withdraw from homesteads mainly includes homesteads and the demolition of the residual value of houses. Local governments generally provide compensation in the form of homesteads, rural apartments, urban commercial housing, and cash [15,16]. Theoretically, the withdrawal from a homestead releases its economic potential constrained by the complex property right structure, and realizes the transformation of rural land from resources to assets [17]. From the perspective of property value realization, the compensation obtained by farmers can directly increase their property incomes and improve their livelihood capital levels [18–20]. However, in practice, the withdrawal modes of rural homesteads neither match the actual needs of the farmers nor the social–economic levels, making it difficult to support farmers' long-term development [21]. If the compensation standards for withdrawal are unreasonable and the monetary compensation amounts are low, they will directly damage the livelihoods of the farmers who withdraw from the homestead. In the survey results of Chen and Ma [22] of Jiangxi Province, the total amount of the CRHW was only RMB 20,000 to 50,000, which is far from enough to build new houses in the local area. The research in Chongqing City also indicates that farmers who withdrew from their homesteads experienced the loss of land income; the main reason is attributed to the low withdrawal compensation [23].

In addition, the social patterns of China's rural areas involve a society of acquaintances centered on individuals, whose influences gradually weaken with the extrapolation of the social network level composed of blood, geography, and occupational relationships [24]. For farmers, withdrawing from the homestead means drastic changes in the spatial, social, and economic environments where they live. If they fail to achieve smooth transitions of their livelihood strategies, farmers will face economic, employment, and social security risks, as well as other aspects [25], which may lead to a decrease in their living standards [26]. Therefore, livelihood diversity and adaptability are critical to the stability of their livelihoods [27,28].

So, how does the CRHW affect the livelihoods of rural households that withdraw from homesteads (especially livelihood capital)? What are the response mechanisms of farmers' livelihoods to the CRHW? What policies should be developed to support the sustainability of farmers' livelihoods? The answers to these questions are crucial to the smooth progress of the rural homestead system reform, rural residents' livelihoods, and social stability. In view of this, we established a theoretical analysis framework and discuss the impacts and mechanisms of the CRHW on the sustainability of livelihoods of farmers in the hope of providing theoretical and practical support for improving the WRH policy and ensuring the farmers' sustainable livelihoods.

The remainder of this paper is organized as follows: Section 2 presents the theory, hypotheses, and models employed in this study. Section 3 describes the data collection, variable selection, and model specification. Section 4 elaborates on the key results of the study. The discussion and conclusion are presented in Sections 5 and 6, respectively.

## 2. Theoretical Framework

The theoretical framework utilized in this study includes three aspects: (1) an analysis of the impact of the CRHW on farmers' sustainable livelihoods; (2) the mediation effect of the adaptation between the CRHW and farmers' sustainable livelihoods; (3) the moderating effect of the livelihood diversity in the relationship between the CRHW and farmers' sustainable livelihoods.

### 2.1. Impact of the CRHW on Farmers' Sustainable Livelihoods

Livelihood is the means of earning a living; it comprises the abilities, assets, and activities required for a way of life. A livelihood is sustainable if it can cope with pressures

and shocks, recover from them, and maintain or enhance its capabilities and assets without damaging any natural resources [29]. The sustainable livelihoods analysis (SLA) framework outlined by the UK Department for International Development (DFID) is widely used in analyzing rural household livelihood issues. This framework is based on the sustainable rural livelihood analysis framework constructed by Scoones [30], which divides people's assets into five types: human capital, natural capital, physical capital, financial capital, and social capital. The establishment of sustainable livelihoods is a process in which people access five types of capital assets, combine and transform those assets to meet their own needs, and finally realize the expansion of livelihood capital [31]. In a sense, the level and allocation of livelihood capital construct the core of sustainable livelihoods [32].

The human capital of farmers mainly includes the labor force, education level, vocational skills, labor ability, health status, and so on. Natural capital refers to the natural resources that farmers rely on and use in the process of establishing and developing their livelihoods, mainly composed of land, water resources, agricultural products, forest products, etc., such as the area of contracted land owned by farmers and the ease of cultivation. Generally speaking, financial capital is the capital reserve, cash flow, and easily realizable equivalent. Deposits, wage income, loans, and insurance all belong to the category of financial capital. Social capital embodies the various social resources that farmers can utilize, such as close relationships with relatives and neighbors, the number of participating social organizations, etc.

Homestead withdrawal is essentially the transfer of land rights owned by farmers [33]. As farmers give up their homesteads, their housing conditions, location conditions, and community environment have changed. These will change farmers' employment patterns, household income compositions, and community relations, thus affecting their livelihood capital accumulations and resulting in different livelihood statuses. Houses/apartments, alternative homesteads, and currencies are the most common types of CRHW [34] that can enhance the livelihood capital directly and promote multi-livelihood capital through the transformation of capitals. Specifically, for farmers who choose houses/apartments or homesteads as compensation, the houses (including houses built on newly arranged homesteads) can directly compensate the physical capital loss of WRH. Monetary compensation can significantly increase financial capital in the short term, and the reasonable allocation of the compensation can lead to synergistic effects to optimize the livelihood capital structures [35]. After relocating to a new concentrated residential community, the cultural and entertainment establishment improvements include more public space; farmers have more opportunities to expand their social networks and enhance their social capital [36]. In addition, services provided by local governments in employment training, social security, and other aspects will also have positive effects in raising farmers' livelihood capital levels. Based on the analysis above, we propose the following hypotheses:

**Hypotheses 1 (H1).** *CRHW can improve farmers' sustainable livelihoods by increasing livelihood capital.*

## 2.2. Mediation Effect of Adaptation

The concept of adaptation originates from studies in evolutionary ecology, and it has been widely used in the fields of climate change [37,38], social sciences [39], and political ecology [40]. In the field of social sciences, adaptation is mostly regarded as the response of human vulnerability or adaptive capacity to the risk of environmental hazards [41].

Withdrawing from a rural homestead is not only a simple spatial location migration, it is also an adaptation process involving cultural, social, and lifestyle changes as well as psychological identification [42]. Farmers who give up their homesteads will gradually adapt to new resettlement sites in terms of behaviors, thinking habits, etc.; only when they psychologically agree can they truly adapt to their lives after relocation and realize the reconstruction and sustainable developments involved in familial livelihoods. From the perspective of local practices, governments have provided basic guarantees for farmers who quit their homesteads, such as medical care, pensions, employment, production,

and life. They also introduce beneficial measures and mobilize resources to provide various vocational skill training for farmers. These measures are designed to eliminate farmers' concerns about housing and living, by expanding farmers' employment channels, eliminating employment barriers, and enhancing their confidence in the future economic situations of their families (to improve their adaptability). Farmers with strong adaptability often have optimistic attitudes. They can actively use their own resources and support policies provided by the government to obtain more employment information, financial support, social relief, and other help so that it is easier to achieve the accumulation and proliferation of livelihood capital. From this point of view, the adaptability of farmers is affected by the CRHW, and to a certain extent, it can also affect their appreciation of livelihood capital and improve their sustainable livelihoods. Based on the above analysis, the following hypothesis is put forward:

**Hypotheses 2 (H2).** *CRHW can improve farmers' sustainable livelihoods by improving their adaptability.*

### 2.3. Moderating Effect of Livelihood Diversity

Livelihood diversity is defined as the process in which rural families establish diverse activity combinations and social support capabilities to survive and improve their living standards [43]. It is the choice of farmers' livelihood strategies, mainly reflected in the diversity of livelihood activities and income sources. Farming households who depend entirely on the agricultural sector are less resilient to risks, and their livelihoods are unsustainable. Therefore, the diversification of livelihood activities is a strategy to minimize risks and uncertainties [44].

Judging from the actual situation in rural China, the types of household livelihood activities mainly include agricultural activities based on planting and breeding, non-agricultural activities based on long-term or temporary employment, and non-agricultural business activities carried out in urban and rural areas. The degree of household livelihood diversity is not only part of the livelihood strategies that farmers choose (actively or passively) based on their livelihood capital, but it also reflects, to a certain extent, the overall utilization of various capitals, the coordination of various resources, and the ability to acquire and use information. For farmers, diversification of livelihood activities is conducive to the rational allocation of urban economic compensation among different livelihood activities, improving the use efficiency of funds and increasing family income. In addition, the government provides a series of support measures, such as vocational training, job promotion, entrepreneurship guidance, loan concessions, etc., for farmers who quit their homesteads. The diversity of household livelihood can effectively transform policy support into upgrading practical vocational skills, employment expansion, entrepreneurial practices, and expanded production, maximizing the utility of non-economic benefits and, thus, improving the level of household livelihood capital. Therefore, in implementing the WRH policy, the impact of the CRHW on the sustainable livelihoods of farmers is also affected by their livelihood diversity to a certain extent. Based on the above analysis, the following hypothesis is put forward:

**Hypotheses 3 (H3).** *Livelihood diversity will positively regulate the relationship between CRHW and farmers' sustainable livelihoods.*

Based on the above analysis, we constructed an analytical framework for the impact of the CRHW, livelihood diversity, and adaptability on sustainable livelihoods (Figure 1).

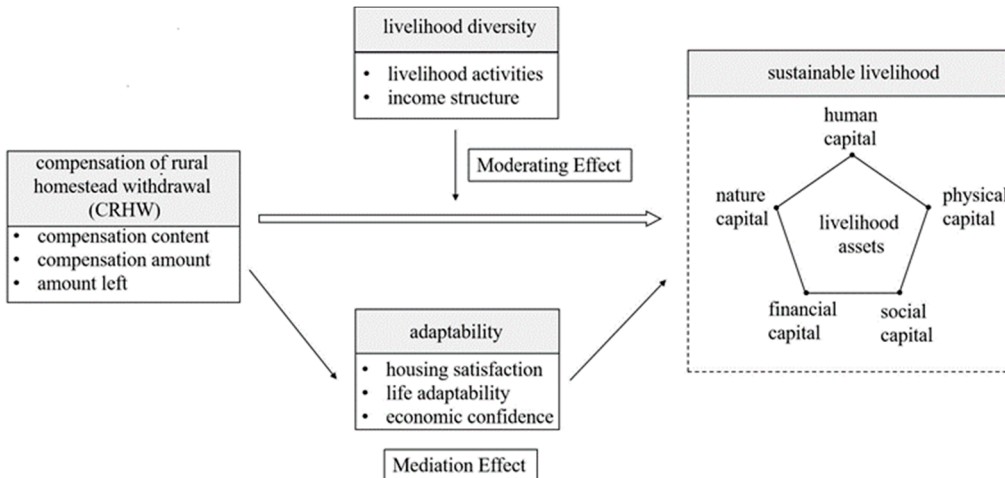

**Figure 1.** Analysis framework of the impact mechanisms of the CRHW, livelihood diversity, and adaptation on sustainable livelihoods.

### 3. Research Data, Variables, and Methods

*3.1. Data Collection*

This study was conducted in Xuzhou City, located in Jiangsu Province, China. Its jurisdiction consists of 5 municipal districts, 3 counties, and 2 county-level cities, with a total area of 11,258 square kilometers and a population of 9.0285 million. Xuzhou's GDP per capita in 2021 was RMB 9361, ranking 11th among 13 cities in Jiangsu Province. Xuzhou is the core city of the "Xuzhou Metropolitan Circle", in which Jiangsu Province focuses on building in the northern region, and is also one of the key areas for the promotion of rural housing improvement projects in northern Jiangsu. The practice of homestead withdrawal in Xuzhou City is representative because it develops the experiences of homestead reforms in the central and southern regions of Jiangsu Province, and reflects the actual situations of farmers' livelihood changes under homestead reforms in the relatively underdeveloped regions of northern Jiangsu. Taking the availability and representativeness of data into account, 5 rural residential communities, which were built and operated in Xuzhou City from 2017 to 2020, were selected for the investigation.

The research data in this article came from a questionnaire survey conducted in November and December 2020. The questionnaire was administered to farmers who withdrew from their homesteads and obtained compensation (e.g., housing and monetary). The content of the questionnaire mainly included three parts: the basic information of the respondents and their families; the livelihood capital of farmers' families, including natural capital, human capital, financial capital, physical capital, and social capital; the data related to the CRHW, including the compensation method, compensation standard, compensation content, compensation amount, etc. Based on random sampling, the research data were obtained through discussions, semi-structured interviews, and questionnaires. A total of 230 valid questionnaires were collected, with an effective rate of 93.9%.

*3.2. Variables*

The dependent variable in this paper was the sustainable livelihoods of farmers, which is represented by 5 types of livelihood capital: human capital, natural capital, physical capital, financial capital, and social capital. Due to the rich connotation of livelihood capital, based on the consideration of data availability and authenticity, with reference to the existing research results [45–47], and in combination with the characteristics of the WRH policy and the actual situation of the surveyed area, the measurement indicators of each livelihood capital were determined. Among them, human capital was measured by 3 indicators, including the number of off-farm employment households, the participation of family labor in skill-training, and the occupational score of off-farm family members [48]; natural capital consisted of 2 indicators: the area of contracted farmland and the distance

between farmers' houses and the farmland; physical capital was measured by 3 indicators: per-capital housing area, categories of durable consumer goods, and categories of public facilities; financial capital was measured by the households' annual income, savings, and the Engel coefficient; social capital was measured by 3 indicators, including whether there were village cadres in the family members, the number of gift expenditures, and entertainment frequency.

The independent variable in this paper is the CRHW obtained by farmers who withdrew from rural homesteads. The CRHW involves a series of compensation and incentive measures taken by the local government for the loss of the homestead, houses, and attachments on the land caused by the farmers participating in the WRH project. The most important CRHW is the withdrawal compensation and the amount of the compensation received by the farmers. According to the survey, the compensation for farmers who withdrew from their homesteads in the study area mainly concentrated on housing and currency. The vast majority of farmers chose to buy houses in the centralized residential area after withdrawing from the homestead. So whether the CRHW could cover a house is an important factor affecting the livelihood of farmers. Therefore, we chose the compensation content number, the amount of the compensation, and the remaining amount of money after purchasing a house in the residential area as the measurement indicators for the CRHW.

In this paper, adaptability, which is selected as the mediating variable, is defined as the response state of the farmers to the reconstruction of the spatial–social–economic environment, and environmental changes in the process of WRH [49]. We focused on armers' psychological adjustment abilities to cope with the external shock of the WRH. Therefore, we selected 3 indicators to evaluate adaptability: housing satisfaction, adaptability of living habits, and confidence in the future economic situation. The adaptability evaluation takes the form of the Likert scale and consists of three statements. According to the attitudes of the respondents to each statement, it was divided into 5 levels, from "very disagree" to "very agree", which are recorded as 1, 2, 3, 4, and 5 points, respectively.

In this paper, livelihood diversity is the moderating variable; according to relevant research results [50], livelihood diversity is defined as the variety of livelihood activities and the diversity of income sources. Therefore, livelihood diversity is measured by the number of livelihood activities and the proportion of non-agricultural income. In the determination of the number of livelihood activities, the livelihood activities of peasant households are divided into 3 types: agricultural activities dominated by agricultural and forestry planting activities and aquaculture activities; off-farm activities based on various forms of short-term and long-term work; operating shops, conducting self-employed activities, such as freight and passenger transport, agricultural product processing, and services. The number of livelihood activities is calculated according to the number of specific categories of activities involved in the livelihoods of farmers.

Previous research provides references to determine the control variables [32,45]. Since the characteristics of the respondents and their families are associated with sustainable livelihoods and refer to the relevant research results, the household size, the highest education level of family members, government support, and family income category is selected as the control variables.

The definition, descriptive statistics, and weight of all variables are shown in Table 1.

**Table 1.** Definition, descriptive statistical analysis, and weight of each variable.

| Variables | Definition | Mean | Std. Dev. | Weights |
|---|---|---|---|---|
| Human capital | | 0.187 | 0.114 | 0.2468 |
| Off-farm employment | Off-farm employment of household members | 1.410 | 0.880 | 0.2928 |
| Skill training | Actual times of the skill training that family members participate in (in survey year): 0 time = 1; 1–2 times = 2; 3–4 times = 3; 5–6 times = 4; 5–6 times = 5. | 1.390 | 0.780 | 0.3577 |
| Occupational score | Occupational score of off-farm employment: employees of private enterprise = 0.25; village cadres = 0.5; employees of state-owned enterprise = 0.75; civil servants = 1 | 0.350 | 0.310 | 0.3495 |
| Natural capital | | 0.596 | 0.103 | 0.1180 |
| Farmland area | Total farmland area (mu) | 4.730 | 2.680 | 0.4440 |
| Farming distance | Distance from the farmer's house to farmland (km) | 1.180 | 1.100 | 0.5560 |
| Physical capital | | 0.399 | 0.096 | 0.1648 |
| Per-capital housing area | Actual housing area per capita in rural areas ($m^2$) | 44.710 | 27.230 | 0.5047 |
| Durable goods | Types of durable goods owned by households: mobile phone, color TV, motorcycle, mobile phone, computer, Internet broadband, refrigerators, air conditioner, washing machine, water heater (species) | 7.730 | 1.540 | 0.1697 |
| Public facilities | Types of public facilities in rural communities (species) | 7.070 | 1.550 | 0.3256 |
| Financial capital | | 0.229 | 0.086 | 0.2251 |
| Annual household income | Annual income of households (RMB $10^4$) | 6.130 | 4.730 | 0.3371 |
| Engel coefficient | Food products shared in total expenditure | 0.400 | 0.200 | 0.2506 |
| Household savings | Actual savings of household (RMB $10^4$) | 1.940 | 3.190 | 0.4123 |
| Social capital | | 0.202 | 0.170 | 0.2453 |
| Village cadres | Whether a family member serves as a rural cadre: no = 0, yes = 1 | 0.110 | 0.310 | 0.3645 |
| Gift expenditure | Annual gift expenditures of households (RMB $10^4$) | 0.600 | 0.580 | 0.3645 |
| Frequency of entertainment | Entertainment activities per month: 0 time = 1; 1–2 times = 2; 3–4 times = 3; 5–6 times = 4; more than 7 times = 5 | 2.300 | 1.590 | 0.2709 |
| CRHW | | 0.417 | 0.206 | — |
| Compensation items | Number of compensation items: housing demolition, land compensation, removal expenses, transition subsidy, loss of working time charges, social subsidy, etc. (number) | 2.935 | 1.318 | 0.3077 |
| Cash compensation | Total amount of cash compensation | 11.214 | 8.240 | 0.4267 |
| Remaining cash | Remaining amount of cash compensation after housing replacement | 0.224 | 6.622 | 0.2656 |
| Livelihood diversity | | 0.551 | 0.147 | |
| Livelihood activities | Number of household livelihood activities (number) | 1.843 | 0.655 | 0.7446 |
| Non-agriculture income ratio | Proportion of non-agriculture income in total income | 0.820 | 0.239 | 0.2554 |
| Adaptability | | 0.825 | 0.211 | |

**Table 1.** *Cont.*

| Variables | Definition | Mean | Std. Dev. | Weights |
|---|---|---|---|---|
| Housing satisfaction | Very satisfied with the housing situation; very disagree = 1; somewhat disagree = 2; neutral = 3; somewhat agree = 4; very agree = 5 | 4.313 | 1.018 | 0.3278 |
| Life adaptability | Adapt to the living habits of the residential communities: very disagree = 1; somewhat disagree = 2; neutral = 3; somewhat agree = 4; very agree = 5 | 4.300 | 0.994 | 0.3340 |
| Confidence | Feel confident in future financial situation: very disagree = 1; somewhat disagree = 2; neutral = 3; somewhat agree = 4; very agree = 5 | 4.291 | 1.073 | 0.3382 |
| Control variable Household size | Number of household members | 3.70 | 1.36 | – |
| Highest educational level | Highest educational levels of family members illiteracy = 1; primary school = 2; middle school = 3; high school = 4; vocational high school/technical school = 5; junior college = 6; undergraduate = 7; postgraduate and above = 8 | 4.20 | 1.38 | – |
| Government support | Belongs to a government-assisted target of aid, such as poor, minimal assurance, or five-guarantee: yes = 0, no = 1 | 0.86 | 0.35 | – |
| Income group * | Income group of household. low income = 1, middle-income = 2, high income = 3 | 2.13 | 0.76 | – |

Note: * According to ratio Z between the disposable incomes of the sample households and the disposable incomes of Xuzhou rural residents in 2017 (RMB 16,697), $Z < 0.5$ are low-income households, $0.5 \leq Z < 1$ are middle-income households, and $Z \geq 1$ are high-income households.

### 3.3. Calculation of Variable Values

The dependent variable, independent variable, mediating variable, and moderating variable in this paper are all indexes that are composed of multiple variables, and the determination of the weights of each index influences the determination of indexes of each variable and the analysis results. The entropy method is a widely used objective assignment method in farmers' livelihood research. It is highly precise, determines the weight of the index according to the degree of dispersion of the original data, and avoids the influence of subjective factors on the research results. Therefore, this paper adopts the entropy method to calculate the weight of each index.

Due to the different characteristics of the variables, and the diversity of data in the dimensions and magnitude, in order to ensure the reliability of the calculation results, this paper uses the standardization method to standardize the original data, uses the entropy value method to determine the weight of each variable, and obtains the value of each comprehensive index. The calculation formula is as follows:

$$S_i = \sum_{j=1}^{m} Y_{ij} w_j \quad (i = 1, 2, \ldots \ldots, n) \tag{1}$$

where $S_i$ is the value of each comprehensive index, including livelihood capital (human capital, natural capital, physical capital, financial capital, social capital), sustainable livelihoods, CRHW, adaptability, and livelihood diversity. $Y_{ij}$ is the value of each indicator after

standardization, and $w_j$ is the weight of each indicator (the weight of each indicator is shown in Table 1).

*3.4. Model Specification*

3.4.1. Logistic Regression Model

Taking *CRHW* as the core explanatory variable and sustainable livelihood as the explained variable, the following regression model is constructed:

$$N_i = \beta_0 + \beta_1 CRHW_i + \beta_2 X_i + \varepsilon_i \tag{2}$$

$N_i$ is the farmers' sustainable livelihoods; $CRHW_i$ is the explanatory variable of homestead withdrawal compensation; $X_i$ represents the control variable affecting the farmers' sustainable livelihoods; $\varepsilon_i$ is the random disturbance term.

3.4.2. Mediating Effect Model

With reference to the research results of Wen Zhonglin et al. (2014) [51], the mediation effect model is constructed:

$$N_i = cCRHW_i + e_1 \tag{3}$$

$$M_i = aCRHW_i + e_2 \tag{4}$$

$$N_i = c'CRHW_i + bM_i + e_3 \tag{5}$$

The coefficient *c* in Formula (3) is the total effect of the independent variable on the dependent variable. The coefficient *a* in Formula (4) is the effect of the independent variable on the mediator ($M_i$). The coefficient *b* in Formula (5) is the direct effect of $M_i$ on the dependent variable after controlling the influence of the independent variable; coefficient *c'* is the direct effect of the independent variable on the dependent variable after controlling the mediators, and $e_1$, $e_2$, and $e_3$ are the regression residuals.

3.4.3. Moderating Effect Model

Referring to the research results of James et al. [52] and Wen Zhonglin et al. [53], the following measurement model is constructed:

$$N_i = \beta_3 + \beta_4 CRHW_i + \beta_5 CRHW_i \times diversity_i + \delta_i \tag{6}$$

The $diversity_i$ refers to the diversity of farmers' livelihoods; $CRHW_i \times diversity_i$ refers to the interaction between the CRHW and livelihood diversity, and $\delta_i$ is the random disturbance term. If the coefficient of $CRHW_i \times diversity_i$ is significant, it indicates that the moderating effect is significant.

## 4. Results

*4.1. Descriptive Statistical Analysis*

The results of the descriptive statistical analysis are presented in Table 2. In general, the sustainability of the farmers' livelihood is poor, the average index of the surveyed farmers is 0.283, which is an extremely low level. In addition, the livelihood capital structure is seriously unbalanced. The maximum of the livelihood capital is natural capital, with a score of 0.596; while the lowest is human capital, only 0.187.

**Table 2.** Impact of the CRHW on sustainable livelihoods and livelihood capital.

| Variable | Sustainable Livelihoods | Human Capital | Natural Capital | Physical Capital | Financial Capital | Social Capital |
|---|---|---|---|---|---|---|
| CRHW | 0.1782 *** | 0.1469 | 0.05541 | 0.1872 *** | 0.0344 | 0.3910 ** |
| | (0.0674) | (0.0935) | (0.9260) | (0.0523) | (0.0911) | (0.1667) |
| Household size | 0.0072 ** | 0.0274 *** | 0.0025 | −0.0261 *** | 0.0159 *** | 0.0045 |
| | (0.0034) | (0.0054) | (0.0059) | (0.0042) | (0.0054) | (0.0085) |
| Highest educational level | 0.0101 *** | 0.0204 *** | −0.0058 | 0.0036 | 0.0148 *** | 0.0085 |
| | (0.0034) | (0.0065) | (0.0055) | (0.0032) | (0.0050) | (0.0084) |
| Government support | 0.0106 | 0.0210 | 0.0122 | 0.0057 | −0.0061 | 0.0303 |
| | (0.0107) | (0.0166) | (0.0139) | (0.0111) | (0.0170) | (0.0285) |
| Income group | 0.0390 *** | 0.0705 *** | 0.0033 | 0.0094 | 0.0641 *** | 0.0250 * |
| | (0.0054) | (0.0092) | (0.0091) | (0.0067) | (0.0091) | (0.0142) |
| Cons | 0.0222 | −0.3576 *** | 0.4168 *** | 0.5039 *** | 0.0253 | −0.1387 |
| | (0.0331) | (0.0628) | (0.0912) | (0.0496) | (0.0514) | (0.0985) |
| $N$ | 230 | 230 | 230 | 230 | 230 | 230 |
| $R^2$ | 0.413 | 0.467 | 0.293 | 0.813 | 0.363 | 0.141 |

Note: * $p < 0.1$, ** $p < 0.05$, *** $p < 0.01$.

Analyzing the compensation obtained by farmers, it was found that the average cash compensation received by farmers was RMB 112,140. However, the remaining amount after purchasing houses in the concentrated residential community was only RMB 22,400. Moreover, nearly half of the farmers received cash compensation that was insufficient to purchase a new house (45.65%). Meanwhile, the farmers' adaptability was generally good after relocating to new regions, with a score of 0.825. More than 80% of respondents felt satisfied with the housing conditions and had confidence in the economic situations of their families. The livelihood diversity was at a medium level, with a score of 0.551. In general, income from non-agricultural employment accounted for more than 80% of total household income, but the number of livelihood activities was only 1.843; the diversity of livelihood needs to improve.

*4.2. Impact of the CRHW on the Sustainable Livelihoods of Farmers*

4.2.1. Logistic Regression Results

Based on the logistic regression model constructed in the early stage, Stata 15.1 software was used to analyze the impact of the CRHW on the sustainable livelihoods and livelihood capital of farmers. The results are shown in Table 2. It can be seen that CRHW helps to improve the sustainable livelihoods of rural households; Hypothesis 1 (H1) was verified. The reasons for the positive impact of the CRHW on sustainable livelihoods were further analyzed. Although the CRHW improved the level of physical capital and social capital of peasant households, the regression results of human capital, natural capital, and financial capital were not significant, which indicates that the improvements in sustainable livelihoods mainly focused on physical and social capital.

In terms of the control variables, the household size, the highest education level of family members, and the income group had significant positive impacts on the sustainable livelihoods, human capital, and financial capital of households. Considering that larger household sizes ensure more labor force, and higher education levels mean a higher quality of labor force, the household size and education level will increase the accumulation of a household's human and financial capital, and improve the sustainable livelihood. The category of household income has a significant positive impact on social capital, this might be because the farmers with higher incomes may spend more on gift expenditures and participate in community cultural and entertainment activities more actively, thus improving their social capital levels.

4.2.2. Endogenous Problems and Estimation Results of the Instrumental Variable Method

The logistic regression results did not take into account the endogenous problems that may exist between the CRHW, the sustainable livelihoods, and livelihood capital. The instrumental variable method was used to solve this endogenous problem. A valid instrumental variable needs to satisfy two conditions: instrument relevance and instrument exogeneity [54], i.e., the instrumental variables should be highly related to the endogenous explanatory variables, but not directly related to the explained variables. Therefore, we chose the original homestead area of farmers and the public transparency of the WRH policy as the instrumental variables. On the one hand, the original homestead area is an important basis for CRHW, which largely determines the compensation amount received by farmers, while public transparency is the basic condition for farmers to participate in the formulation of the homestead withdrawal policy and affect the CRHW. Therefore, both satisfy the correlation condition. On the other hand, these two terms do not have a direct impact on the current livelihood capital of farmers and satisfy the exogenous conditions. Therefore, theoretically, the original homestead area and the public transparency of policies can be used as instrumental variables.

Two-stage least squares (2SLS) estimation was performed using the instrumental variables, and the results are shown in Table 3. After using the original homestead area and the public transparency of policy as tool variables, the CRHW still has a significant positive impact on the sustainable livelihoods, physical capital, and social capital of the peasant households, which is consistent with the logistic regression model. The results demonstrate that the tool variables selected in this study were robust, and further verify the robustness of the logistic regression model.

**Table 3.** Instrumental variable regression analysis: 2SLS.

| Variable | Sustainable Livelihoods | Physical Capital | Social Capital |
|---|---|---|---|
| CRHW | 0.6452 ** (0.2833) | 0.5460 ** (0.2477) | 1.5217 ** (0.7093) |
| Household size | −0.0006 (0.0064) | −0.0321 *** (0.0057) | −0.0144 (0.0155) |
| Highest educational level | 0.0119 *** (0.0041) | 0.0050 (0.0036) | 0.0130 (0.0099) |
| Government support | 0.0066 (0.0133) | 0.0026 (0.0126) | 0.0207 (0.0344) |
| Income group | 0.0403 *** (0.0064) | 0.0104 (0.0069) | 0.0281 * (0.0169) |
| Cons | −0.1023 (0.0854) | 0.4083 *** (0.0858) | −0.4401 * (0.2134) |
| $N$ | 230 | 230 | 230 |
| $R^2$ | 121.83 | 1184.10 | 35.83 |

Note: * $p < 0.1$, ** $p < 0.05$, *** $p < 0.01$.

*4.3. Mechanism Analysis of the Sustainable Livelihood Response to the CRHW*

4.3.1. Mediating Effect of Adaptability

Based on the mediating effect model that was constructed, the mediating effect of adaptability was tested with reference to previous research [51]. The results are shown in Table 4. CRHW has a significant positive impact on sustainable livelihoods and adaptability, with regression coefficients of 0.1782 and 0.3040, respectively. That is, the higher the CRHW, the stronger the sustainable livelihoods and adaptability of the peasant households. Moreover, the regression coefficients of adaptability and sustainable livelihoods were significantly positive, which means the improvement of household adaptability was conducive to promoting the sustainable livelihood level. After the mediating variable

was added, the regression coefficient of the CRHW and sustainable livelihoods were still significantly positive, indicating that adaptability plays a partial intermediary role in the CRHW and sustainable livelihoods. The proportion of the mediating effect to the total effect was 7.25% [1]; Hypothesis 2 (H2) was verified. That is, the improvement of the CRHW will directly raise the sustainable livelihoods of rural households. At the same time, the CRHW can indirectly improve sustainable livelihoods by enhancing adaptability. The intermediary effect of adaptability accounts for 7.25%.

**Table 4.** Mediation analysis of adaptability between the CRHW and sustainable livelihoods.

| Variable | Sustainable Livelihoods | Adaptability | Sustainable Livelihoods | Sustainable Livelihoods |
|---|---|---|---|---|
| CRHW | 0.1782 *** (0.0674) | 0.3040 *** (0.0890) | – – | 0.1610 *** (0.0494) |
| Adaptability | – – | – – | 0.0595 ** (0.0232) | 0.0425 * (0.0226) |
| Household size | 0.0072 ** (0.0034) | −0.0022 (0.0117) | 0.0113 *** (0.0033) | 0.0082 ** (0.0034) |
| Highest educational level | 0.0101 *** (0.0034) | 0.0112 (0.0114) | 0.0093 ** (0.0037) | 0.0088 *** (0.0033) |
| Government support | 0.0106 (0.0107) | 0.0507 (0.0432) | 0.0107 (0.0117) | 0.0081 (0.0110) |
| Income group | 0.0390 *** (0.0054) | 0.0383 * (0.0198) | 0.0399 *** (0.0055) | 0.0379 *** (0.0054) |
| Cons | 0.0222 (0.0331) | 0.4370 * (0.228) | 0.0232 (0.0262) | 0.0044 (0.0274) |
| *N* | 230 | 230 | 230 | 230 |
| $R^2$ | 0.413 | 0.122 | 0.331 | 0.385 |

Note: * $p < 0.1$, ** $p < 0.05$, *** $p < 0.01$.

Next, we examined whether adaptability played a role in mediating the process of the CRHW promoting the farmers' physical and social capital; the results are shown in Table 5. After adding the intermediary variable, the regression coefficient of the CRHW and the physical capital is still significantly positive, which is the same for the regression coefficient of the CRHW and the social capital. In the meantime, the regression coefficients of adaptability and social capital were significant, indicating that adaptability played a partial mediating role between the CRHW and social capital, and the proportion of the mediating effect to the total effect was 9.41% [2]. Since the regression coefficients of adaptability and physical capital were not significant, the bootstrap method was adopted to confirm the mediating effect of adaptability.

According to the mediation effect analysis program proposed by Zhao [55], the number of repeated samples was set to 500 and the confidence interval was set to 95%, and the two confidence intervals of bias correction and percentile were used for the estimation. The analysis results are shown in Table 6. The confidence of the percentile and bias-corrected intervals were between −0.0211 and 0.0357 and 0.0193 and 0.0388, respectively, both of which include 0, indicating that the mediating effect of adaptability was not significant. After controlling the intermediary variable of the adaptability, CRHW had a significant positive impact on the physical capital (percentile = −0.4720~−0.3473, bias-corrected = −0.4753~−0.3508; both excluding 0). Therefore, CRHW had a direct effect on the physical capital of the peasant households, and the intermediary role of adaptability did not exist.

**Table 5.** Mediation analysis of adaptability between the CRHW and physical and social capital.

| Variable | Physical Capital | Physical Capital | Physical Capital | Social Capital | Social Capital | Social Capital |
|---|---|---|---|---|---|---|
| CRHW | 0.1872 *** (0.0523) | – – | 0.1840 *** (0.0517) | 0.3910 ** (0.1667) | – – | 0.3390 *** (0.1060) |
| Adaptability | – – | −0.0033 (0.0378) | −0.0159 (0.0327) | – – | 0.1280 ** (0.0593) | 0.1210 ** (0.0557) |
| Household size | −0.0261 *** (0.0042) | −0.0201 *** (0.00483 | −0.0260 *** (0.0042) | 0.0045 (0.0085) | 0.0109 (0.0086) | 0.0051 (0.0084) |
| Highest educational level | 0.0036 (0.0032) | −0.0007 (0.0045) | 0.0037 (0.0032) | 0.0085 (0.0084) | 0.0062 (0.0084) | 0.0061 (0.0081) |
| Government support | 0.0057 (0.0111) | 0.0038 (0.0210) | 0.0064 (0.0115) | 0.0303 (0.0285) | 0.0276 (0.0291) | 0.0247 (0.0277) |
| Income group | 0.0094 (0.0067) | 0.0052 (0.0090) | 0.0098 (0.0065) | 0.0250 * (0.0142) | 0.0200 (0.0142) | 0.0215 (0.0140) |
| Cons | 0.5039 *** (0.0496) | 0.572 *** (0.0611) | 0.5140 *** (0.0542) | −0.1387 (0.0985) | −0.1060 (0.0884) | −0.1870 ** (0.0838) |
| N | 230 | 230 | 230 | 230 | 230 | 230 |
| $R^2$ | 0.813 | 0.522 | 0.814 | 0.141 | 0.128 | 0.153 |

Note: * $p < 0.1$, ** $p < 0.05$, *** $p < 0.01$.

**Table 6.** Mediation analysis of adaptability between CRHW and physical capital: bootstrap method.

| Category | Observed Coef. | Bias | Bootstrap Std. Err. | 95% Conf. Interval | | | |
|---|---|---|---|---|---|---|---|
| | | | | Percentile | | Bias-Corrected | |
| Direct effect | −0.4090 | 0.0027 | 0.0309 | −0.4720 | −0.3473 | −0.4753 | −0.3508 |
| Mediating effect | 0.0064 | −0.0004 | 0.0145 | −0.0211 | 0.0357 | −0.0193 | 0.0388 |

### 4.3.2. Regulating Effect of Livelihood Diversity

According to the regulatory effect model constructed in the early stage and referring to the research of Wen [53], we tested whether livelihood diversity played a regulatory role in the CRHW affecting sustainable livelihoods. The results are shown in Table 7. The regression coefficient of "CRHW × diversity" and sustainable livelihoods was 0.2985, which is significant at the level of 0.1, indicating that the impact of the CRHW on sustainable livelihoods was regulated by livelihood diversity. When the degree of livelihood diversity was higher, the positive impact of the CRHW on sustainable livelihoods was stronger, and Hypothesis 3 (H3) was verified. Moreover, the regulatory role of livelihood diversity in the CRHW and the livelihood capital were tested. According to the same judgment method, the positive impact of the CRHW on social capital was regulated by livelihood diversity, i.e., the higher the degree of livelihood diversity of the peasant households, the stronger the positive impact of the CRHW on the social capital. The promoting effect of the CRHW on the physical capital was not regulated by livelihood diversity.

**Table 7.** Moderation analysis of livelihood diversity between the CRHW, sustainable livelihoods, and livelihood capital.

| Variable | SL | PC | SC |
|---|---|---|---|
| CRHW × diversity | 0.2985 * (0.1543) | 0.1195 (0.1601) | 0.8113 ** (0.3764) |
| CRHW | 0.0037 (0.0954) | 0.1229 (0.1065) | −0.0862 (0.2357) |
| livelihood diversity | −0.1168 * (0.0707) | −0.1343 * (0.0811) | −0.2697 (0.1828) |
| Household size | 0.0068 * (0.0035) | −0.0238 (0.0039) | 0.0022 (0.0087) |
| Highest educational level | 0.0099 *** (0.0035) | 0.0047 (0.0031) | 0.0074 (0.0085) |
| Government support | 0.0121 (0.0110) | 0.0090 (0.0115) | 0.0328 (0.0293) |
| Income group | 0.0370 *** (0.0061) | 0.0151 ** (0.0066) | 0.0160 (0.0153) |
| Cons | 0.0897 ** (0.0439) | 0.5302 *** (0.0679) | 0.0451 (0.1187) |
| N | 230 | 230 | 230 |
| $R^2$ | 0.4244 | 0.4639 | 0.1613 |

Note: * $p < 0.1$, ** $p < 0.05$, *** $p < 0.01$.

## 5. Discussion

The results show that the CRHW has a positive impact on farmers' sustainable livelihoods. This view is consistent with previous studies, i.e., after the farmers withdraw from their homesteads, the total amount of livelihood capital increased [33,56] and the welfare level improved [57,58]. The main reason for the enhancement of farmers' sustainable livelihoods lies in that the CRHW significantly promotes the levels of material and social capital. After withdrawing from the homestead and moving into concentrated residential communities, farmers owned more durable goods and were around convenient public infrastructure, which led to an increase in material capital. Moreover, new residences are often equipped with complete cultural and recreational facilities. The frequency of farmers' participation in public activities increased significantly, and their relationships with the communities became closer, thus increasing their social capital.

At the same time, the CRHW has not increased the farmers' financial capital, which is inconsistent with the proposition that homestead withdrawal can increase farmers' property incomes [59]. On the one hand, the cost of living increases after farmers move to concentrated residential communities. The main reason is that the production land around the house (for planting vegetables and raising poultry) has greatly reduced, and farmers' expenditures on food have increased. Meanwhile, farmers' energy use patterns have changed, and expenditures on drinking water and gas have increased. On the other hand, monetary compensation may be unable to cover the costs of farmers' relocating. After withdrawing from their homesteads, most farmers bought apartments in concentrated residential communities, decorated houses, and purchased furniture and household appliances. Farmers' spending often exceeds the compensation they receive. In addition, human capital is the most important capital type in the livelihood capital of rural households and is the key to improving the overall livelihood capital level and optimizing the livelihood capital structure [60]. The development of labor ability via skill training is an important way for farmers to accumulate human capital. However, owing to the migration of rural male labor, the rural permanent population is mainly composed of those who are middle-aged, elderly, and women [61,62]. Although some areas provide employment training and employment recommendation opportunities, they are often unsuitable for the farmers' conditions and

demands. It is difficult for farmers to obtain effective support to improve human capital and enhance livelihood diversity and adaptability.

In addition, the compensation standards were considered low and unsatisfactory by most farmers. There is an academic consensus that farmers' homestead withdrawal incomes are lower than theoretical expectations [63,64]. In China, the rural homestead is subject to "three rights division" [65]: the village collective enjoys the ownership, while the farmers obtain the qualification right by virtue of their collective membership, and they share the right of use. However, farmers, as important rights holders of homesteads, obtain the least income when it comes to land occupation. According to an estimation, in the sales income of land acquisition, 60–70% is taken by county–township governments, 25% by village collectives, and only 5–10% by farmers [66]. Compared with rural land acquisition, when farmers are persuaded to withdraw from the homestead, they face more unitary, lower compensations. Therefore, one may wonder if farmers' livelihoods are sustainable when formulating compensation policies [67].

## 6. Conclusions and Policy Implications

This paper establishes an analytical framework for the impact of the CRHW and adaptability/livelihood diversity on farmers' sustainable livelihoods. We conducted an empirical test using survey data from Xuzhou City, Jiangsu Province. The results identified that CRHW improves the sustainable livelihoods of farmers and increases their physical and social capital. The instrumental variable estimation also confirms the robustness of the results. In addition, the mechanism test shows that CRHW can improve farmers' sustainable livelihoods and social capital by enhancing their adaptability. Moreover, the impact of the CRHW on farmers' sustainable livelihoods and social capital is regulated by livelihood diversity. This study can be a reference for rural land system reforms in other developing countries under the background of urbanization.

Based on the findings of this study, the following policy recommendations are proposed: (1) Conduct a comprehensive survey on the livelihoods of the households of farmers to promote the implementation of the WRH policy. The WRH policy is an enormous shock to the livelihoods of rural households. Farmers have to face the loss of livelihood capital caused by giving up their homesteads and adjust their livelihood strategies to make a living. Thus, it is necessary to conduct a comprehensive survey before the implementation of WRH, obtain the livelihood capital situation, and compensation willingness of farmers to formulate targeted compensation policies. (2) Develop reasonable compensation standards of the CRHW to guarantee farmers' land rights and interests. The standard land price of a homestead is an important basis for determining the compensation standards of the CRHW. For local governments, professional institutions should be hired to formulate the benchmark land prices of rural homesteads. Moreover, a reasonable compensation standard of the CRHW should fully consider the value of homesteads to farmers' livelihood capital accumulation, for the purpose of supporting farmers' sustainable livelihoods. (3) Adopt a variety of compensation methods to improve the adaptability and livelihood diversity of farmers. Regarding housing replacement, homestead replacement, and cash compensation, the incomes of farmers mainly depend on agriculture, which aggravates their livelihood vulnerabilities. Therefore, a social security system is expected to be gradually built for farmers who withdraw from rural homesteads, covering medical security, pension security, employment security, children's education security, minimum living security, etc., to help farmers adapt to their lives after relocation. Moreover, it is essential to broaden the income channels of farmers and increase their economic incomes by providing technical training, so as to improve the diversities of their familial livelihoods.

**Author Contributions:** Conceptualization, W.Q., Z.L. and C.Y.; formal analysis, W.Q.; data curation, W.Q.; writing—original draft preparation, W.Q.; writing—review and editing, W.Q., Z.L. and C.Y.; supervision, Z.L. and C.Y.; funding acquisition, Z.L. All authors have read and agreed to the published version of the manuscript.

**Funding:** This research was funded by the soft science research project of the Rural Revitalization Expert Advisory Committee of the Ministry of agriculture and rural affairs of the Central Agricultural Office of the People's Republic of China "tracking research on the pilot of rural homestead system reform (rkx202025c)".

**Data Availability Statement:** The data, models, and code used for the research reported in this paper are available from the corresponding author upon reasonable request.

**Conflicts of Interest:** The authors declare no conflict of interest.

## Notes

[1]　According to the mediating effect model constructed above, the calculation formula is: $a \times b/c = (0.3040 \times 0.0425)/0.1782 = 7.25\%$.

[2]　According to the mediating effect model constructed above, the calculation formula is: $a \times b/c = (0.3040 \times 0.1210)/0.3910 = 9.41\%$.

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
