# Peer review of "Response Mechanism of Farmers’ Livelihood Capital to the Compensation for Rural Homestead Withdrawal—Empirical Evidence from Xuzhou City, China"

_land, doi:10.3390/land11122149_

Round 1

Reviewer 1 Report

This is a meaningful study. Based on the primary survey data from Xuzhou, the authors studies the impact of CRHW on the sustainable livelihood of farmers and its impact mechanism. A few comments for your reference:

1. The sustainable livelihood analysis framework includes livelihood capital, livelihood strategy and livelihood results. Generally speaking, the term “livelihoods” refers to livelihood results, not livelihood capital. Therefore, according to the content of the manuscript, it is suggested that the title “Livelihoods” be replaced by “Livelihoods Capital” in the manuscript title.

2. In Hypothesis 1, whether and how CRHW affects the livelihood capital of farmers, the author does not clarify the internal mechanism.

3. Line 288-291, the interpretation of the formula needs to be modified

4. The instrumental variable “original homestead area of farmers” does not meet the theoretical assumptions.

5. The author has poorly discussed the results of the paper. One would expect to find the previous empirical work enriching the discussions of the results, but unfortunately, that has not been done.

6. The policy recommendations needs to be further improved, and it is suggested that the author further expand the policy based on the core research results.

Reviewer 2 Report

The article is clear, easy to follow and logically structured. The structure of the model is clear, easy to construct and well described. The argumentation and methodology in the paper is convincing. However, the references need be checked, eg. change Zhang Z to Zhang C.

Author Response

Point 1: The article is clear, easy to follow and logically structured. The structure of the model is clear, easy to construct and well described. The argumentation and methodology in the paper is convincing. However, the references need be checked, eg. change Zhang Z to Zhang C.

Response 1: Thanks for this comment. We have checked and revised all the references according to your suggestion.

Author Response

Point 1: There are many grammatical errors throughout the MS sections, they are too enormous to correct one by one in the detailed comments. Please consider revise them.

Response 1: Thanks for this comment to make this manuscript clearer for readers. We have carefully checked and made correction according to your suggestion. And the whole manuscript have been polished by a colleagues working in the United States.

Point 2: The resolution of the figure is low making it hard to read, please add higher resolution figure

Response 2: Thanks for this comment. We have provided a figure with higher resolution for easy reading.

Point 3: Does the collected data come from questioner or online survey? Not clear!!!!

Response 3: Thanks for this comment. The research data in this article comes from a questionnaire survey administered as a face-to-face interview by the researcher. We have added description of data collection in 3.1 more in detail.

Point 4: Tables 2-5: It is recommended to spell out the variables not acronym.

Response 4: Thanks for this comment. According to your suggestion, we have replaced the abbreviations of variables with full names in table 2-5.

Point 5: what are the values between prentices?

Response 5: Thanks for this comment. In previous studies, there have been different opinions on the impact of homestead withdrawal on farmers' livelihood [1-4]. This paper establishes an analytical framework for the impact of CRHW, adaptability and livelihood diversity on farmers' sustainable livelihoods, and conducts an empirical test using survey data of Xuzhou City, Jiangsu Province, in the hope of providing a theoretical and practical support for improving the WRH policy and ensuring the farmers’ sustainable livelihoods. Besides, this study can provide reference for the rural land system reform in other developing countries under the background of urbanization.

Related Reference:

1. Jia, H.; Wang S. Analysis on the Welfare Changes of Farmers before and after Concentrated Residence and Its Influencing Factors - - Based on the Survey of Farmers in Jiangsu Province. China Rural Survey 2014, (01), 26-39+80.

2. Liu, C.; Wang, K.; Ou, M. Study on the welfare level of farmers’exiting from homestead and living in concentration from the perspective of farmers’  Resources and Environment in the Yangtze Basin. 2020, 29(03), 748-757.

3. Yang, L.; Zhu, C.; Yuan S.; Li, S. Analysis on farmers’willingness to rural residential land exit and welfare change based on the supply-side reform: A case of Yiwu City in Zhejiang Province. China Land Science. 2018, 32(01), 35-41.

4. Lang, X. Multiple paths to realize farmers' homestead property rights under the system of ' separation of three rights'. Academics2022, (02),146-155.

Round 2

Reviewer 1 Report

Many changes have been made to the manuscript, but spell-check is needed.

For example, "necessry" should be changed to "necessary", "conducte" should be changed to "conduct" in Line 516, etc.

Author Response

Point 1:

Many changes have been made to the manuscript, but spell-check is needed.

For example, "necessry" should be changed to "necessary", "conducte" should be changed to "conduct" in Line 516, etc.

Response 1: 

Thanks for this comment. We are very sorry for the spelling mistakes in the manuscript. We have examined the full manuscript to correct spelling and grammatical errors.